# Should the Use of Patient Medical Information in Research Require the Approval of Attending Physicians?

**Eisuke Nakazawa** [1], **Shoichi Maeda** [2,3], **Makoto Udagawa** [1] and **Akira Akabayashi** [1,4,*]

1   Department of Biomedical Ethics, Faculty of Medicine, University of Tokyo, 7-3-1 Hongo, Bunkyo-ku, Tokyo 113-0033, Japan
2   Course for Health Care Management, Graduate School of Health Management, Keio University, Fujisawa 252-0883, Japan
3   Department of Health Policy and Management, School of Medicine, Keio University, 35 Shinjuku-ku, Tokyo 160-8582, Japan
4   Division of Medical Ethics, New York University School of Medicine, 227 East 30th Street, New York, NY 10016, USA
*   Correspondence: akira.akabayashi@gmail.com or akirasan-tky@umin.ac.jp; Tel.: +81-3-5841-3511; Fax: +81-3-5841-3319

**Abstract:** Retrospective observational studies using medical records require researchers to guarantee the right to opt out of the study. However, is it also necessary to confirm whether the medical professionals who created those medical records permit their use as well? In this article, we consider possible options based on a fictitious scenario. Based on our deliberations, we recommend that the information be disclosed on the hospital's homepage or in leaflets (principal investigator: hospital director), and, similar to patients, attending physicians should be given the opportunity to opt out. We also recommend that an application be submitted to the hospital's research ethics committee. In this paper, we address the public interest aspect of the use of patient information as a primary item for ethical scrutiny. In addition to research ethics, this particular point underscores the importance of public health ethics, particularly as they pertain to the conflict between individual freedom and public interest.

**Keywords:** publication ethics; authorship; retrospective study; informed consent; medical records; attending physician

## 1. Introduction

With the recent increase in the popularity of the large-scale use of retrospective patient data in research, the utility of patient information has been recognized. In retrospective observational studies using medical records, patients must be guaranteed the right to opt out of the study. However, is it also necessary to confirm whether the medical professionals who created those medical records approve their use as well? In this article, we deliberate on the necessity of consent from medical professionals when using patient information for research purposes.

*Scenario: In a large hospital with 1500 beds, a research plan using the medical record information of all patients in an anonymous fashion was proposed. The principal investigator of this study is Hospital Director A. For patients, the information regarding the research plan is disclosed on both the hospital's homepage (HP) and on in-hospital posters, and the opportunity to opt out is guaranteed for those who refuse to participate in the study.*

*Seeing this proposal, Professor B from Department C lodges a complaint to A, stating that he/she would not want his/her patients' information to be used without permission, even if the information is anonymized and unlinkable. In response, A maintains that the proposal is perfectly acceptable because it was approved by the hospital director (A) on the basis of*

*the fact that medical records belong to the hospital and that their use for research purposes has been appropriately disclosed to the patients, who have also been given the opportunity to opt out. B is not convinced and brings this matter to the attention of other medical staff members during a department meeting. Assistant Professor C, who is in charge of the outpatient clinic of the hospital, agrees with B, insisting that they would also prefer that the information of certain (several) patients not to be used for research purposes. As medical records can be seen by all of the healthcare professionals related to the cases within them, this is not a breach of privacy. From the perspective of confidentiality obligations, the information can be shared among related healthcare professionals.*

This method of obtaining consent from patients (opt-out) is specific to Japan. On hospital home pages (HP), researchers upload documents providing notice that patient records will be used and that the data will be anonymized, as well as explaining the aim of research. Additionally, the researchers write down their contact addresses for patients who do not want to participate in the research. More specifically, patients who refuse to have their data used by the researchers, even when anonymized, can opt-out. The justifications for this method are that (1) it guarantees the patients' right to refuse to participate in the research and (2) the document clearly states that the patients do not suffer any consequences if they opt-out. This method has been criticized because there is no guarantee that patients will see the HP, and many patients may have moved to different hospitals or may have already died. Family members could act as surrogates for these patients, but it is hard to imagine family members of deceased patients checking the hospital HP. Accordingly, this method is valid for obtaining informed consent, and the ethics of it are questionable. However, this method is officially permitted according to governmental guidelines, such as in the "Ethical Guidelines for Medical and Biological Research Involving Human Subjects" [1].

Whether medical records belong to hospitals or patients is still a topic of active discussion [2] and has generated many different perspectives and opinions. Some argue that the concept of ownership does not apply to data because data are not physical objects; in fact, they can be simultaneously owned by multiple people and can even be replicated. Others have debated the ownership rights of the medical information included in a medical chart (i.e., does it belong to the physician who prepared it or to the patient who received the medical examinations?). This is entirely a moot point. There is no problem in saying that medical data belong to both physicians and patients; the more important matter is who is in charge of managing the information and who should be allowed to access the information as well as the conditions under which access should be granted. Though not extensive, there is some legal debate concerning the rights of the attending physician (the author of the medical records) with regard to medical treatment information. However, these debates are conducted by considering the provision of medical treatment information to patients and not situations in which it is provided to physicians and researchers for research purposes [3].

Legal arguments are beyond the scope of this paper. Instead, our discussion focuses on specific and practical solutions that can better address situations such as the above scenario and that also encompass the relevant ethical perspectives on this matter. Another point is the crucial difference between quality evaluations of hospital activities and research is that evaluation activities are essential to the functioning of a hospital, but no specific research study is essential for the hospital to function. If a physician opts out of having their patients' data used in research, researchers may be able to provide other data to the hospital that enable the hospital to evaluate its activities.

## 2. Discussion

Considering the title question, "should the use of patient medical information in research require the approval of attending physicians", it may be useful to organize the possible options using the corollary framework of obtaining consent from patients. Currently, researchers who conduct studies that use information concerning a large number of

patients at facilities such as hospitals often proceed by "disclosing information on HP and introducing opt-out consent to guarantee the right of patients to refuse participation." As stated above, many have questioned whether this method is truly accessible to patients and whether approval obtained in this way is considered valid, among other concerns. Nevertheless, this approach is currently accepted by research ethics committees (REC), albeit as a suboptimal measure. Given the above, what are the options that are available in the scenario described above?

1.  Obtain oral or written consent from all healthcare professionals related to the cases and submit an application to the REC.
2.  Obtain consent from all department heads orally or in writing and submit an application to the REC.
3.  Disclose information on the hospital's HP or in leaflets (principal investigator: hospital director) and guarantee the medical professionals in the hospital the opportunity to opt-out if they do not approve and submit an application to the REC.
4.  Since medical records belong to the hospital, it should suffice to have the person responsible for those records (the hospital director) approve their use and to ensure that an application that addresses patient opt-out possibilities is submitted to seek approval from the REC, as is the case for normal applications.
5.  Since medical records belong to the hospital, it is up to the hospital director to decide how they should be used; there is no need to consider the intentions of individual medical professionals. The hospital director only needs to deal with patients who are opting out. There is no need for an application to be submitted to the REC.

Immensely helpful when considering this issue are the descriptions in the "Guidance for the Proper Handling of Personal Information by Medical and Nursing Care Providers." These descriptions have been provided in the guidelines by the Ministry of Health, Labour and Welfare that are presented below [4].

> For example, medical records include not only data from a patient's objective examinations but also from a physician's judgment and assessment made on the basis of the patient's data. These contents are all considered the patient's personal information, but from the perspective of the physician who created the medical records, their notes on their judgement and assessment are also considered information relating to the individual physician. Therefore, it should be kept in mind that medical records contain some information that is regarded as both the personal information of the patient as well as that of the physician, i.e., the dual nature of medical records. (p.6, emphasis added by the authors)

In light of the interpretation provided above, option one can be easily eliminated, as it totally ignores the aspects of considering the physician and the REC. For the same reason, option four (i.e., application to the REC only) would also not lead to approval. Meanwhile, option one is unrealistic given the low feasibility of the hospital director (principal investigator) contacting a large number of attending physicians to ask for consent to use patient records. There is also the possibility that some of the physicians who created the original medical records have been transferred to other institutions. Option two may be feasible in terms of the number of personnel involved, but department heads cannot possibly represent the intentions of all of the medical staff in their departments. Asking each staff member about their intentions is impractical. Moreover, if coming from a superior, such an order might coerce subordinate medical staff into responding in a way that is not truly reflective of their intentions. Based on these ethical considerations, option three is the only remaining option. This option involves the same exact steps that are taken for patients (i.e., required information disclosure on HP and the required provision of an opt-out choice). The relative strength of this option would be that, if it is disclosed internally (e.g., via the HP), there would likely be plenty of opportunities for physicians to view this information and spread the word. On the other hand, one drawback is that, even when the commonly used phrasing "refusal to consent would not cause any disadvantages" is used, refusal

by "opting out" may entail tangible or intangible sanctions. This point is unique to the physician's scenario. Although personal information such as a physician's notes on their judgements and assessments made on the basis of a patient's data is important to protect (it is a physician's right), it is not absolute, and this is the same for patients. At present, however, the authors have not come up with any practical solution other than option three to proceed with a study requiring approval from the REC.

Finally, we want to express our stance in this unique paper. A lot of research ethics papers stress the importance of protecting patient safety and preventing the leakage of private patient information, namely autonomy or self-determination, or, in a classical sense, risk–benefit analysis. Our analysis is the first to (1) protect private patient information and (2) to protect physician interpretations made at the time of assessment when considering patient data. These interpretations might include a physician's original way of thinking. For example, in a case report, the physician's interpretation of patient data would be evaluated as having a high level of originality and could lead to new assessment/evaluation methods for diagnosing patients. Additionally, in psychiatric cases, the interpretations made by physicians are quite important to further treatment and psychotherapy approaches. Patients have the right to access their own medical records. However, considerations have to be made regarding the effects that physician notes may have on a patient if, for example, a physician writes that a patient is schizophrenic and has no chance of recovering; if they write that a patient with depression is at high risk of committing suicide, even if the patient did not mention the possibility of committing suicide; or if they write that "although this adjustment disorder patient says that the patients have a trouble at the workplace, but from the intake regarding family history, I strongly suspect that this patient's spouse seems to be a cheating spouse and having extramarital affairs, thus the patients is suffering." Thus, two sets of medical records may be needed. In this manner, this paper is unique in that it deals with very practical and important issues in a real-world clinical setting. We have focused on these very practical and difficult issues in this paper.

## 3. Recommendations and Summary

The authors—all of whom are REC members with experience in research ethics review propose that, at present, it is best to adopt option three when pursuing research studies using medical records in Japan. (Option 3: Disclose information on the hospital's HP or in leaflets [principal investigator: hospital director] and guarantee the medical professionals in the hospital the opportunity to opt-out if they do not approve and submit an application to the REC.) Additionally, its conclusions are not limited by the Japanese setting. We welcome opinions and suggestions from our readers.

One major ethical issue concerns the public interest aspect of patient information [5]. This is a core issue of public health ethics [6] that finds individual human rights (freedom) to be in opposition to the public interest of society. For instance, should patients cooperate in research in urgent and meaningful areas such as COVID-19 (public interest), or should the freedom of individuals be respected in making that decision? Although this issue is beyond the scope of our present considerations, it warrants deeper discussion from ethical as well as legal perspectives in the future.

**Author Contributions:** Conceptualization, E.N. and A.A.; methodology, E.N. and S.M.; validation, E.N. and M.U.; formal analysis, E.N. and S.M.; writing—original draft preparation, E.N.; writing—review and editing, E.N., S.M., M.U. and A.A.; project administration, A.A. All authors have read and agreed to the published version of the manuscript.

**Funding:** This research received no external funding.

**Data Availability Statement:** Not applicable.

**Acknowledgments:** The authors thank Tomohide Ibuki from the Tokyo University of Science, Japan, for his insightful comments on the manuscript.

**Conflicts of Interest:** The authors declare no conflict of interest.

**Disclaimer:** This paper presents personal views of the authors, which do not represent the position of the REC to which the authors belong.

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
