# Peer review of "Should the Use of Patient Medical Information in Research Require the Approval of Attending Physicians?"

_publications, doi:10.3390/publications10030027_

Round 1

Reviewer 1 Report

Review of “Should research use of patients’ medical information require approval of attending physicians?”

This brief report introduces and explores an interesting ethical issue pertaining to research access to patients’ medical records. The fictional case and its analysis are set in a hospital in Japan and refer to Japanese policies, laws, and procedures. However, the issue is relevant to virtually any medical setting, and its conclusions are not limited by this Japanese setting.

The authors’ argument rests on an insight found in Japanese legal guidance for the use of medical information: The data physicians collect and record about their patients also reveals something about them as clinicians. In addition to the results of laboratory tests, the medical record reveals which tests were ordered and when. In addition to the findings of a physical exam, the record includes their interpretation, the diagnosis, the plan of care, and so forth. They key point is that a patient’s medical record includes implicit information about that person’s physicians, and so those physicians may also have concerns about their own privacy, confidentiality, and even a stake in data ownership. This, the authors argue, gives physicians the ethical and legal standing to have those medical records excluded from retrospective research studies. The authors propose modifying informed consent procedures for such studies to allow physicians to opt out of having their patients’ records included.

This article raises a novel issue and raises interesting questions. I recommend that it be accepted for publication with some modifications.

The scenario used in this paper would be more useful if there were a little more detail about Professor B’s objections to the study. Is B objecting in order to protect their patients? If so, what unwarranted risk does B see in the proposed study? Is B objecting because they claim (legal or moral) authority over accessing their patients’ records? Perhaps B is objecting because the informed consent process (for patients) is inadequate. Is B concerned that an analysis of their patients’ records might reveal that B is not providing good medical care (or is deficient in maintaining complete medical records)? It is difficult to assess this situation without a fuller understanding of it. The authors’ analysis of this situation suggests that B’s objection is based on a kind of ownership claim – B gathered and recorded the information in the patient’s records, and in doing so made various choices and judgments that weave B’s clinical skills into the fabric of the patient’s medical record. It would help to make this clearer.

The authors consider how patients’ medical data is used in retrospective research studies. It would be helpful to add a brief paragraph about whether their conclusions would also apply to the use of medical records for non-research purposes such as quality assessment. Allowing physicians to have data about their patients excluded from a quality assessment analysis would seem to defeat the purpose of that activity. So, perhaps the authors could explain why research and quality assessment differ in their access to patients’ records. (A suggestion: Perhaps the crucial difference is that quality assessment activities are essential to the functioning of a hospital, but no specific research study is essential. If a physician opts out of having their patients’ data used in research, the researchers may be able to find other data to fill that gap. But if a physician were allowed to opt out of having their patient’s records used in hospital quality assessment programs, the results of those assessments would be seriously compromised.)

Suppose a researcher designs a retrospective study of the effectiveness of a particular treatment for a particular condition. This is announced on the hospital’s HP, and patients are given the opportunity to opt out. Some patients may be very interested in this project, and want to contribute by having their data included – but their physician objects to having their data used. Should the physician be able to override the patient’s choice in this matter? The current system in Japan allows patients to opt out of having their data included in research, but makes no provision for having them express positive consent to research participation. As a practical matter, the physician’s objection to having their patients’ data used in a study would override an individual patient’s desire to be included in that study. Further consideration is needed regarding whether this warrants large-scale changes to the consent system.

Author Response

Review 1 of “Should research use of patients’ medical information require approval of attending physicians?”

This brief report introduces and explores an interesting ethical issue pertaining to research access to patients’ medical records. The fictional case and its analysis are set in a hospital in Japan and refer to Japanese policies, laws, and procedures. However, the issue is relevant to virtually any medical setting, and its conclusions are not limited by this Japanese setting.

Thank you for your comment. We have amended the conclusion as follows:

The authors—all of whom are REC members with experience in research ethics review—propose that, at present, it is best to adopt Option 3 in pursuing research studies using medical records in Japan. Additionally, its conclusions are not limited by the Japanese setting.

The authors’ argument rests on an insight found in Japanese legal guidance for the use of medical information: The data physicians collect and record about their patients also reveals something about them as clinicians. In addition to the results of laboratory tests, the medical record reveals which tests were ordered and when. In addition to the findings of a physical exam, the record includes their interpretation, the diagnosis, the plan of care, and so forth.

They key point is that a patient’s medical record includes implicit information about that

person’s physicians, and so those physicians may also have concerns about their own privacy, confidentiality, and even a stake in data ownership. This, the authors argue, gives physicians the ethical and legal standing to have those medical records excluded from retrospective research studies. The authors propose modifying informed consent procedures for such studies to allow physicians to opt out of having their patients’ records included.

This article raises a novel issue and raises interesting questions. I recommend that it be accepted for publication with some modifications.

The scenario used in this paper would be more useful if there were a little more detail about Professor B’s objections to the study. Is B objecting in order to protect their patients? If so, what unwarranted risk does B see in the proposed study? Is B objecting because they claim

(legal or moral) authority over accessing their patients’ records? Perhaps B is objecting because the informed consent process (for patients) is inadequate. Is B concerned that an analysis of their patients’ records might reveal that B is not providing good medical care (or is deficient in maintaining complete medical records)? It is difficult to assess this situation without a fuller understanding of it. The authors’ analysis of this situation suggests that B’s objection is based on a kind of ownership claim – B gathered and recorded the information in the patient’s

records, and in doing so made various choices and judgments that weave B’s clinical skills into the fabric of the patient’s medical record. It would help to make this clearer.

Thank you for your feedback. You have already listed all possibilities.

Is B objecting in order to protect their patients? If so, what unwarranted risk does B see in the proposed study? 

⇒ The risk of being able to identify the individual patient cannot be completely excluded. Total genome analysis data, although anonymized, could still identify the patients.

Perhaps B is objecting because the informed consent process (for patients) is inadequate.

⇒ This might be a possibility, as medical malpractice suits are more common lately.

B gathered and recorded the information in the patient’s records, and in doing so made various choices and judgments that weave B’s clinical skills into the fabric of the patient’s medical record.

⇒ This might also be a possibility.

Is B concerned that an analysis of their patients’ records might reveal that B is not providing good medical care (or is deficient in maintaining complete medical records)?

⇒ This might also be a possibility.

The authors’ analysis of this situation suggests that B’s objection is based on a kind of ownership claim

⇒ As you correctly point out, our main concern is ownership. What you mentioned is exactly what we want to assert.

The data physicians collect and record about their patients also reveals something about them as clinicians. In addition to the results of laboratory tests, the medical record reveals which tests were ordered and when. In addition to the findings of a physical exam, the record includes their interpretation, the diagnosis, the plan of care, and so forth.

They key point is that a patient’s medical record includes implicit information about that

person’s physicians, and so those physicians may also have concerns about their own privacy, confidentiality, and even a stake in data ownership.

I would also like to touch on the issue of “tiered medical records,” which I am sure you are well aware of. This debate can be traced back to the U.S. 25 years ago, when Westin (1997) proposed a two-tiered system of medical records, which was based on the idea that the autonomic disclosure of medical records may cause problems for both patients and health care providers. The first-level records contain what is considered to be an official record, which includes family history, symptoms, test results, diagnosis, treatment plans, medication prescriptions, payment records, etc. Patients have full access to this part of the record. On the other hand, the second-level records contain sensitive judgments about the patient’s emotional and psychological state, as well as speculative and tentative hypotheses that the physician wishes to keep for personal purposes. If the patient requests access to these second-level records, a procedure is followed to grant them access. However, this occurs when the patient’s trust in their physician has dissipated.

If such a system is in place, we believe that all first-level records could be used for research.

Westin AF. Medical records: Should patients have access?  Hastings Center Rep 7:23-

28, 1977.

The authors consider how patients’ medical data is used in retrospective research studies. It would be helpful to add a brief paragraph about whether their conclusions would also apply to the use of medical records for non-research purposes such as quality assessment. Allowing physicians to have data about their patients excluded from a quality assessment analysis would seem to defeat the purpose of that activity. So, perhaps the authors could explain why research and quality assessment differ in their access to patients’ records. (A suggestion: Perhaps the crucial difference is that quality assessment activities are essential to the functioning of a hospital, but no specific research study is essential. If a physician opts out of having their

patients’ data used in research, the researchers may be able to find other data to fill that gap.

Thank you for your comment. In line with your suggestion, we added the following paragraph: “[T]he crucial difference between quality assessment and research is that quality assessment activities are essential to the functioning of a hospital, but no specific research study is essential. If a physician opts out of having their patients’ data used in research, the researchers may be able to find other data to fill that gap.”

But if a physician were allowed to opt out of having their patient’s records used in hospital quality assessment programs, the results of those assessments would be seriously compromised.

Yes, we agree.

Suppose a researcher designs a retrospective study of the effectiveness of a particular

treatment for a particular condition. This is announced on the hospital’s HP, and patients are given the opportunity to opt out. Some patients may be very interested in this project, and want to contribute by having their data included – but their physician objects to having their data used. Should the physician be able to override the patient’s choice in this matter? The current system in Japan allows patients to opt out of having their data included in research, but makes no provision for having them express positive consent to research participation. As a

practical matter, the physician’s objection to having their patients’ data used in a study would override an individual patient’s desire to be included in that study. Further consideration is needed regarding whether this warrants large-scale changes to the consent system.

You are quite right. Therefore, the important issue concerns the ownership of the medical records. This is why we have a legal expert among our collaborators. The truth is that there is no clear legal answer to this in Japan.

In Japan, the Medical Practitioners Law (Article 24, Paragraph 1 and Article 24, Paragraph 2) stipulates the obligation to record and preserve medical records. In other words, when a physician provides medical treatment, they must immediately record the details of the treatment in the patient’s medical record (Article 24, Paragraph 1). In addition, the administrator of the hospital or clinic (or the physician in case the physician treats patients at home, etc., as a private individual) must preserve the records for five years (Article 24, Paragraph 2).

As mentioned above, the Medical Practitioners Law imposes on the administrator of a medical institution the obligation to preserve medical records. However, it does not specify who the information contained in the medical records (not only patient information in the narrow sense, but also information contained in the medical records, including the physician’s evaluation of the patient’s treatment) should belong to.

In Japan, the Act on the Protection of Personal Information stipulates that the disclosure of such information is possible. Thus, patients can request medical institutions to disclose their own medical information.

Therefore, it seems that all parties involved in a patient's medical record (patient, healthcare professionals, and hospital) have a certain amount of ownership over the record. Thus, a situation where a patient agrees to allow their medical records to be used for research but their physician disagrees was not contemplated. Ethically, if the patient’s self-determination is given priority, the physician would comply with their wishes after informing them of the risks (e.g., information leakage). If the physician’s wishes are to be respected, then respect for their right to self-determination would prevail, since this would serve to protection their own information. As one would expect, there would be no case in which a hospital refuses a patient's wish, because such a situation would make it impossible to conduct research. I hope this sufficiently answers your query.

We are incredibly grateful for your valuable comments.  We hope you find our responses satisfactory.

Author Response

Review 2 Report: Should Research Use of Patients’ Medical Information Require Approval of Attending Physicians?

Summary: In this brief paper, the authors consider an important dilemma that arises in public health research: though patients’ ability to opt-out of (having their information used in) a study is the subject of much work in public health and research ethics, what’s less-often scrutinized is whether medical professionals’ consent should be required. That is, the authors ask whether medical professionals should, like patients themselves, have the ability to opt out of having their patients’ medical records and the data therein used in retrospective observational studies. They ultimately conclude that the most feasible and ethically sound possibility is for research governance bodies to “Disclose information on the hospital’s [homepage] or in leaflets…and give [this information to] medical professionals in the hospital concerned, [and] guarantee the opportunity to opt-out if they do not approve” prior to submitting the relevant information to the hospital’s research ethics committee.

This paper deals with questions including: (1) Setting aside legal questions of intellectual property rights, whose interests–patients’, physicians’, hospital administrators’, etc.--should be weighted most heavily in deciding how to use medical information? (2) Is opt-out consent really consent, given that we might be justified in assuming that most patients will not actually read the ‘terms and conditions,’ so to speak, of their treatment?

General comments:

  • **all page numbers will refer to the page number given on the upper right-hand corner of the pdf**

⇒Thank you for pointing this out. We have changed the location of the page numbering.

  • I really enjoyed reading this paper. I think the authors do a great job of drawing readers’ attention to the “dual nature of medical records,” (p. 4) which is often overlooked in favor of an almost-exclusive focus on patients’ privacy rights. Understanding medical records in this dual way really helps to establish that they’re dialogical, and that the physicians and other medical professionals who contribute to such records aren’t mere collectors of information, but are themselves active participants in the creation of that information.

⇒Thank you very much.

  • The pragmatic force of this paper is also quite strong, as the authors compellingly make the case for why ethicists should be concerned about medical professionals’ ability to opt-out of having their patients’ data used in a study.

⇒Thank you very much.

  • One thing that I wondered while reading is whether there might be noteworthy differences between the types of consent that medical professionals (as opposed to patients) can actually

There may be two patterns. 1) Consent to use all the data; and 2) Consent to use the ‘certain portion’ of the medical record.

⇒To answer your query, I would like to touch on the issue of “tiered medical records,” which I am sure you are well aware of. This debate can be traced back to the U.S. 25 years ago, when Westin (1997) proposed a two-tiered system of medical records, which was based on the idea that the autonomic disclosure of medical records may cause problems for both patients and health care providers. The first-level records contain what is considered to be an official record, which includes family history, symptoms, test results, diagnosis, treatment plans, medication prescriptions, payment records, etc. Patients have full access to this part of the record. On the other hand, the second-level records contain sensitive judgments about the patient’s emotional and psychological state, as well as speculative and tentative hypotheses that the physician wishes to keep for personal purposes. If the patient requests access to these second-level records, a procedure is followed to grant them access. However, this occurs when the patient’s trust in their physician has dissipated.

If such a system is in place, we believe that all first-level records could be used for research.

Westin AF. Medical records: Should patients have access?  Hastings Center Rep 7:23-28, 1977.

  • Moreover, something that seems to be hinted at but not fully addressed is the different reasons that physicians might have for not wanting particular patients’ data included in a research sample.

According to Westin, such information tends to be speculative, tentative hypotheses, etc., which the medical practitioner prefer to keep to themselves.

Specific points that might warrant further clarification:

  • One thing that might read as a bit vague is the mention of “medical professionals in the hospital concerned” in option 3 on page 3.

  • First, I think that, semantically, there are two different readings of this sentence, and I wasn’t entirely sure which was the authors’ intention. Is the sentence

focusing on the concerned medical professionals in the hospital, or more generally on the medical professionals in the concerned hospital? In other words, it isn’t totally clear to me what “concerned” is modifying here.

We meant the former. ‘Concerned’ is misleading, so we have altered the phrase accordingly:

Disclose information on the hospital’s HP or in leaflets (principal investigator: hospital director) and guarantee the medical professionals in the hospital the opportunity to opt-out if they do not approve.

  • The second, less nitpicky (ha) concern that might be raised here is the scope of the medical professionals Are we concerned primarily with attending physicians? It seems like the answer to this is no, given the difference between options 1 and 3. But still, are we concerned with attending physicians and residents/interns? What about registered nurses, or nurse practitioners, or physicians’ assistants? Furthermore, what about LPNs/CNAs? I think that my question here can be more succinctly phrased as: How do we determine which medical professionals have a vested concern in a particular ‘piece’ of patient data?

Thank you for your feedback. We are not concerned only with attending physicians. Registered nurses, nurse practitioners, physicians’ assistants, and LPNs/CNAs are all related health care professionals.

Accordingly, we have changed the phrase of option 1 as follows:

Obtain oral or written consent from all healthcare professionals related to the cases. Submit an application to the REC.

  • With respect to the latter point, it’s also entirely possible that my own situatedness in the USA is coloring my reading here – it’s possible that, simply by mentioning a specific geographical region in the italicized example on pages 2-3, this concern could be skirted, as I admittedly quite unfamiliar with who is in patient-facing roles in other parts of the world, including Japan. Or, the authors could avoid this concern by using language more specific than ‘medical professionals,’ perhaps referring exclusively to physicians, or something to that

Thank you for your comment. We have changed the phrase as follows: Medical record can be seen by all healthcare professionals related to the cases. This is not a breach of privacy. From the perspective of the confidentiality obligation, the information can be shared within the related healthcare professionals .

  • I’m also a bit curious about the authors’ claim on p. 4 that only physicians need be concerned that “refusal by ‘opting out’ may entail tangible or intangible sanctions.” On the one hand, this makes perfect sense, given that the authors have already mentioned that the scope of their paper is retrospective observational studies, so the patients’ data has already (presumably) been collected. However, there might be some warranted concern about long-term patients, or those who will return for some further/other treatment in the future, and whether having opted out of having their data included in research studies could feasibly worsen the quality of care they Of course, this is quite the hypothetical, and the patients’ concerns would still be markedly different from those of a physician who opts out of including his patients’ data in a study.

Thank you for your feedback. What we wanted to convey through the statement is:

One drawback is that, even with the stereotypical phrase “refusal to consent would not cause any disadvantages,” refusal by “opting out” may entail tangible or intangible sanctions. This point is unique to the physician’s scenario.

If many physicians refuse consent or opt-out, the quality of the research data may diminish. Hospital director A (principal investigator) and the researchers involved in that retrospective study would feel uncomfortable because their data’s value will decrease, Unlike in the case of patients, sanctions can easily be delivered within the hospital’s narrow hierarchy (i.e., Hospital Director A or other related researchers can easily spread rumors.) Therefore, this point is unique to the physician’s scenario.  

  • Lastly, also on p. 4, I’m wondering whether public health ethics necessarily “finds individual human rights (freedom) to be in opposition with the public interest of society.” Though I think the two are quite often at odds, and though this may be unduly optimistic of me, I’m not sure that there isn’t any way in which we can view a commitment to public interest as compatible with an expansive understanding of personal freedom.

Your comment is very insightful. We should not always reduce the public ethical problem to the conflict between individual freedoms and public interests. We are always thinking about how we can make people recognize “medical records” as public property. The strong societal sense of individualism can hopefully be weakened in the case of medical records, since medical care is provided through national resources.

Overall, I really enjoyed this paper, and look forward to seeing how it develops!

⇒ We are incredibly grateful for your valuable comments. We tried our best to improve the manuscript based on your feedback, and we hope you find the revised manuscript to be suitable for publication in your journal.

Round 2

Reviewer 2 Report

This looks great! I think that some of the edits you've all made have really helped make the 'framing' of the paper more clear, and have helped to curtail some of the concerns I raised in an earlier review. A few of the points of concern raised in the initial review weren't addressed, but I don't think this necessarily detracts from the paper's quality! 

A couple spots that I think could be clarified, maybe even just through tweaking the wording, include: (1) "Although the personal information of the physicians..." on p. 4, (2) I had trouble understanding how the second highlighted bit on p. 3 ("The crucial difference...") ties in; I think this might be aided by even more straightforwardly saying what you mean by 'quality assessment.' 

Author Response

Response to Reviewer 2

This looks great! I think that some of the edits you've all made have really helped make the 'framing' of the paper more clear, and have helped to curtail some of the concerns I raised in an earlier review. A few of the points of concern raised in the initial review weren't addressed, but I don't think this necessarily detracts from the paper's quality! 

A couple spots that I think could be clarified, maybe even just through tweaking the wording, include:

  • "Although the personal information of the physicians..." on p. 4,

⇒ We have changed to ‘[A]lthough personal information such as a physician’s notes on their judgements and assessments made on the basis of a patient’s data.’

  • I had trouble understanding how the second highlighted bit on p. 3 ("The crucial difference...") ties in; I think this might be aided by even more straightforwardly saying what you mean by 'quality assessment.

⇒The crucial difference between quality assessment and research is that quality assessment activities are essential to the functioning of a hospital, but no specific research study is essential. If a physician opts out of having their patients’ data used in research, the researchers may be able to find other data to fill that gap.

We have changed to:

Th crucial difference between quality evaluations of hospital activities and research is that evaluation activities are essential to the functioning of a hospital, but no specific research study is essential for the hospital to function. If a physician opts out of having their patients’ data used in research, researchers may be able to provide other data to the hospital that enable the hospital to evaluate its activities.

Once again, thank you for reading our manuscript. I hope this manuscript is now publishable to your journal.
